# Diagnosis and Management of Simple and Complicated Meconium Ileus in Cystic Fibrosis, a Systematic Review

**DOI:** 10.3390/diagnostics14111179

**Published:** 2024-06-04

**Authors:** Mădălina Andreea Donos, Gabriela Ghiga, Laura Mihaela Trandafir, Elena Cojocaru, Viorel Țarcă, Lăcrămioara Ionela Butnariu, Valentin Bernic, Eugenia Moroșan, Iulia Cristina Roca, Dana Elena Mîndru, Elena Țarcă

**Affiliations:** 1Saint Mary Emergency Hospital for Children, 700309 Iasi, Romania; madalina.donos@umfiasi.ro; 2Pediatrics Department, “Grigore T. Popa” University of Medicine and Pharmacy, 700115 Iasi, Romania; mindru.dana@umfiasi.ro; 3Department of Morphofunctional Sciences I—Pathology, “Grigore T. Popa” University of Medicine and Pharmacy, 700115 Iasi, Romania; elena2.cojocaru@umfiasi.ro (E.C.); eugenia_morosan@yahoo.com (E.M.); 4Department of Preventive Medicine and Interdisciplinarity, “Grigore T. Popa” University of Medicine and Pharmacy, 700115 Iasi, Romania; viorel.tarca@umfiasi.ro; 5Department of Medical Genetics, Faculty of Medicine, “Grigore T. Popa” University of Medicine and Pharmacy, 700115 Iasi, Romania; ionela.butnariu@umfiasi.ro; 6Department of Surgery II, “Saint Spiridon” Hospital, 700115 Iasi, Romania; bernicvalik@yahoo.com; 7Department of Surgery II, “Grigore T. Popa” University of Medicine and Pharmacy, 700115 Iasi, Romania; iulia.roca@umfiasi.ro; 8Department of Surgery II—Pediatric Surgery, “Grigore T. Popa” University of Medicine and Pharmacy, 700115 Iasi, Romania; tarca.elena@umfiasi.ro

**Keywords:** meconium ileus, cystic fibrosis, diagnostic, management, neonates

## Abstract

The early management of neonates with meconium ileus (MI) and cystic fibrosis (CF) is highly variable across countries and is not standardized. We conducted a systematic review according to the Preferred Reporting Items for Systematic Reviews and Meta-analyses statement. The protocol was registered in PROSPERO (CRD42024522838). Studies from three providers of academic search engines were checked for inclusion criteria, using the following search terms: meconium ileus AND cystic fibrosis OR mucoviscidosis. Regarding the patient population studied, the inclusion criteria were defined using our predefined PICOT framework: studies on neonates with simple or complicated meconium which were confirmed to have cystic fibrosis and were conservatively managed or surgically treated. Results: A total of 566 publications from the last 10 years were verified by the authors of this review to find the most recent and relevant data, and only 8 met the inclusion criteria. Prenatally diagnosed meconium pseudocysts, bowel dilation, and ascites on ultrasound are predictors of neonatal surgery and risk factor for negative 12-month clinical outcomes in MI-CF newborns. For simple MI, conservative treatment with hypertonic solutions enemas can be effective in more than 25% of cases. If repeated enemas fail to disimpact the bowels, the Bishop–Koop stoma is a safe option. No comprehensive research has been conducted so far to determine the ideal surgical protocol for complicated MI. We only found three studies that reported the types of stomas performed and another study comparing the outcomes of patients depending on the surgical management; the conclusions are contradictory especially since the number of cases analyzed in each study was small. Between 18% and 38% of patients with complicated MI will require reoperation for various complications and the mortality rate varies between 0% and 8%. Conclusion: This study reveals a lack of strong data to support management decisions, unequivocally shows that the care of infants with MI is not standardized, and suggests a great need for international collaborative studies.

## 1. Introduction

Cystic fibrosis (CF) is an autosomal recessive disease caused by a defect in a gene on the long arm of Chromosome 7 which codes for the cystic fibrosis transmembrane conductance regulator (CFTR), a chloride channel on epithelial surfaces [1]. Among the thousands of mutations that can be involved, the best known is *delta F508*, being responsible for unregulated epithelial sodium channels (ENaCs), decreased chloride secretion, and increased sodium resorption on epithelial surfaces. Other known mutations responsible for the appearance of CF manifestations are *G542X*, *W1282X*, *R553X*, *G551D*, and other modifier genes, all explaining approximately 17% of the phenotypic variability [2]. The risk for a patient to present with meconium ileus (MI) is 24.9% if he has two copies of *F508del* mutation, 16.9% if he has one *F508del* paired with another mutation and 12.5% for two other CFTR mutations [3]. While individual studies have found various modifier genes that either increase or decrease the risk of MI, there has been limited capacity to reproduce these findings [4,5,6].

CFTR is in charge of the excretion of both Cl^–^ and HCO_3_^−^ in the small intestine. The tight matrix of exocytosed mucins in the gut lumen is mediated by HCO_3_^−^, which is essential for chelating Ca^2+^ to generate normal, loose, well-hydrated mucus [2]. Abnormal HCO_3_^−^ secretion decreases luminal pH and creates an acidic, dehydrated environment and thick mucus. Increased mineral content, protein-bound carbohydrates, and greater amounts of stool albumin are further encouraged by the excessively acidic luminal environment. Together with the viscous mucus, these result in viscid meconium, which physically obstructs the terminal ileum [3].

Meconium ileus occurs in between 12% and 20% of neonates diagnosed with CF and is usually the first manifestation of the disease; it is defined as a mechanical small bowel obstruction in the perinatal period, caused by inspissated meconium within the terminal ileum [7]. When there is only the occlusion created by the thickened meconium, meconium ileus is classified as “simple”. In cases with antenatal manifestation, MI can coexist with meconial peritonitis, intestinal atresia, perforations, and necrosis or ileal volvulus; in this case, MI is “complicated”. In neonates with CF, simple and complex MI happen at comparable rates [3]. The early management of neonates with MI is highly variable across countries and is not standardized [8]. Also, the optimal surgical technique remains controversial, and because primary anastomosis may result in complication rates between 21% and 31%, ileostomy and delayed anastomosis is recommended by some authors [2,9], while others recommend primary anastomosis to avoid electrolyte losses and the need for surgical reintervention to close the stoma [10].

This article sets out to explore a difficult topic that completely lacks standardization, diagnosis, and management of simple and complicated meconium ileus in cystic fibrosis. The aim of our review is to summarize the latest data from the specialized literature regarding the treatment of simple and complicated meconium ileus, in the hope of discovering and applying uniform treatment norms.

## 2. Materials and Methods

This systematic review was conducted according to the Preferred Reporting Items for Systematic Reviews and Meta-analyses (PRISMA) statement [11]. The protocol was registered in PROSPERO under the Registration Number: CRD42024522838.

### 2.1. Search Strategy of Electronic Databases

Between 15 March 2024 and 30 April 2024 we performed a systematic review of the data available with regard to the management of MI in cystic fibrosis. We evaluated articles from three providers of academic search engines (PubMed, EMBASE, and Web of Science) using the following search terms: meconium ileus AND cystic fibrosis OR mucoviscidosis. We then used a combination of MESH terms like neonatal intestinal obstruction OR meconial peritonitis OR constipation OR distal intestinal obstruction syndrome AND cystic fibrosis OR mucoviscidosis to find the most relevant articles about the diagnosis and management of this neonatal occlusion syndrome. 

#### 2.1.1. Including and Excluding Criteria

Studies included in this review met the following criteria: full-text available online, published in the last 10 years, clearly stated methodology, human subjects, and articles available in English. We searched the reference lists of the included articles in order to identify other potentially relevant articles. We excluded the duplicated, non-relevant studies and the ones treating non-human subjects; after the title and abstract assessment a total of 112 studies were excluded.

Regarding the patient population studied, the inclusion criteria were defined using our predefined PICOT framework: studies on neonates with simple or complicated meconium who were confirmed to have cystic fibrosis and were conservatively managed or surgically treated. These two groups were compared for the need of surgery and placement of an ileostomy (main outcomes). The secondary studied outcomes were the imagistic findings, the number of days until weaning from total parenteral nutrition (TPN), days of hospitalization, the number of days until stoma closure, and mortality rate. 

#### 2.1.2. Selection of Studies and Information Extraction

Three authors [M.A.D., E.T., and L.M.T.] independently selected the articles included in the review by searching databases. They independently screened all titles and abstracts of identified studies for eligibility. If disagreement between the reviewers existed, consensus was formed or a fourth reviewer [G.G.] acted as referee. Duplicate references and non-human studies were eliminated. Title and abstract assessment were performed by V.T., E.C. and L.I.B. References review of the relevant articles were reviewed by V.B. and D.E.M. for pertinent articles that would meet inclusion and exclusion criteria; 112 articles were initially selected, then excluded by E.M. and I.C.R. as being non-relevant. 

## 3. Results

From Web of Science, 890 articles were found discussing meconium ileus AND cystic fibrosis OR mucoviscidosis; 63,111 were selected from PubMed and 89,676 from EMBASE. After applying the filters (full-text available online, published in the last 10 years, English language, humans, age: birth-1-month, publication types: articles, reviews, meta-analyses), 454 studies remained to be screened for eligibility. During the second step, we reviewed the titles, abstracts, and the full texts of the papers and selected the studies reporting neonatal intestinal obstruction OR meconial peritonitis OR constipation OR surgery to extract pertinent data for our review (Figure 1). A total of 566 publications were verified by the authors of this review to find the most recent and relevant data, and only 8 met the inclusion criteria (Table 1). Forty-seven relevant studies with their citations were included in our reference list.

### 3.1. Prenatal Diagnosis and Fetal Risk Factors

An intrauterine growth restriction and prenatal ultrasound findings including hyperechoic masses (signifying inspissated stool in the distal ileum), peritoneal calcifications, or non-visualization of the gallbladder may suggest CF in a high-risk fetus [2,17]. Fetal meconium is hypoechoic or isoechoic to surrounding abdominal structures in the second and third trimesters, whereas MI manifests as a hyperechoic mass with sonographic density higher than that of the liver or bone. If the prenatal ultrasound is abnormal, the parents’ carrier status should be determined using a common mutation panel or entire CFTR gene sequencing. Adequate genetic counseling must be provided to address the risks of having a child with CF and its potential consequences if both parents are carriers of the disease. Then, the fetus should undergo ultrasound monitoring every six weeks and the delivery should be scheduled at a tertiary care facility with a multidisciplinary team including an experienced perinatologist, a pediatric surgeon, and a well-equipped neonatal intensive care unit (NICU). One will expect a premature birth and a reduced birth weight; the reduced birth weight of neonates with cystic fibrosis (CF) is mostly due to reduced intrauterine growth, which is only partially accounted for by shorter gestations [18].

In the last 10 years, we only found three original studies addressing the issue of prenatal ultrasound findings and the impact on postnatal management of neonates with MI and CF. In 2020, Shinar et al. analyzed a case series and performed a meta-analysis for identifying prenatal predictors of neonatal surgery; they identified 24 patients with MI and CF from 244 MI studied patients and found that meconium pseudocysts, bowel dilation, and ascites are prenatal predictors of neonatal surgery in cases of meconium peritonitis. Their postoperative mortality rate was 8.1% and the survival rate 100% for neonates not requiring surgery [12]. Padoan et al., in a retrospective, multicenter, observational study on 71 patients, also found that prenatally diagnosed intestinal obstruction is a risk factor for negative 12-month clinical outcomes in MI-CF newborns [13]. 

Jessula et al. in 2018 also performed a retrospective cohort review of prospectively collected data in order to identify if specific genetic subtypes and the presence of echogenic bowel on prenatal US are associated with either MI or operative requirement in the neonatal period. They identified seven patients with simple MI, who were treated conservatively and did not require any surgical intervention, three patients with simple MI, and sixteen with complicated MI who were operated. Prematurity and LBW were risk factors for MI and the need for surgery. Specific genotypes and echogenic bowel were not predictors of either [15]. 

### 3.2. Clinical Presentation and Diagnosis of Meconium Ileus

Increased sodium resorption through unregulated ENaCs is accompanied by water resorption and results in dehydrated mucus in the lungs and luminal contents of the small bowel, beginning from intrauterine life. Due to the deficiency in pancreatic enzymes and abnormal mucin production the meconium is thickened, protein-rich, and inspissated in the distal ileum, causing intrinsic obstruction and dilation of the small intestine upstream, until its perforation with meconium peritonitis. Approximately 15–20% of fetuses with cystic fibrosis manifest from birth with simple or complicated meconial ileus with meconium peritonitis. Although the clinical, ultrasound, and radiological appearance is different for simple versus complicated meconium ileus, the definitive diagnosis of cystic fibrosis is established based on genetic testing or biochemical alterations of the CFTR. In assessing the infant’s clinical condition, a laboratory assessment that considers lactate, hemoglobin, white blood cell counts (WBC), and electrolytes is also needed [3,19]. 

The method for screening newborns for CF is usually predicated on two assessments of immunoreactive trypsinogen levels detected in a dried blood spot on the Guthrie card, the first test in the days 2–5 of life and the second being carried out within 30 days of birth [20]. This algorithm has a rather high false-positive result rate; also, the diagnosis is not ruled out in the event of a negative newborn screening result [Smith]. High b-IRT level was detected as a risk factor for negative 12-month clinical outcomes in MI-CF newborns in the study conducted by Padoan et al. in 2019 [13]. Sweat testing is performed to confirm or rule out cystic fibrosis if screening results are positive. In cases where the newborn is well-hydrated and not edematous, a sweat chloride test can be performed as soon as 48 h after delivery. This test is diagnostic at concentrations higher than 60 mmol/L and represents the gold standard biochemical analysis for CF. Nonetheless, many children diagnosed with MI require primary genetic testing because they are either too little, underweight, edematous, or too sick to undergo sweat testing right away.

CFTR functional testing and the identification of two mutations linked to cystic fibrosis are additional methods of diagnosis. A specific set of mutant probes can be used to evaluate one of the 40 most prevalent mutations seen in over 90% of mucoviscidosis patients. Anyway, this method is less sensitive than an expanded gene analysis, and is less likely to reveal polymorphisms and mutations of unknown significance [21].

#### 3.2.1. Simple Meconium Ileus

Simple MI manifests from birth with the impossibility of eliminating meconium, bilious vomiting, and distended abdomen, with dilated intestinal loops visible on the abdominal wall (Figure 2). 

These clinical manifestations will raise the suspicion of cystic fibrosis and investigations will be continued with thoracoabdominal X-ray, ultrasound, genetic investigations, and sweat test. The ultrasound may show pseudo-thickening of the bowel walls because of thick and adhesive meconium. Stratified and dehydrated meconium causes the bowel lining to have increased echo intensity. Intestinal gases cannot be eliminated due to the high density of meconium and its adhesion to the intestinal walls, so that the abdomen becomes intensely weathered, occlusive, but in the absence of air–fluid levels. The abdominal radiograph has almost the same appearance in the supine and erect position with distended loops without air–fluid levels and a ground glass or ‘soap-bubble’ appearance (Neuhauser’s sign) (Figure 3). Another typical occurrence on X-ray is inspissated pellets along the large intestine wall. If there is a total obstruction, there might not be any air in the rectum.

It could be acceptable to take blood and urine cultures and think about starting broad spectrum antibiotic medication if there is a fever or an elevated white blood cell count [22]. A nasogastric tube is implanted to facilitate the decompression of the stomach and proximal small bowel, to stop more bilious emesis and lower the risk of aspiration. Neonates with simple MI may achieve successful decompression after noninvasive management with Gastrografin or other contrast repeated enemas performed under fluoroscopic or ultrasound guidance, the success rates varying between 5% and 83% [2,8]. According to some authors, the percentage of babies who are treated without surgery has been declining over time [8,23]. In the last 10 years, we only found five studies reporting a total number of 133 cases of simple MI [8,9,10,13,15]; from them, 70 patients received at least one enema, with a success rate of 54% (38 patients responded to enemas and did not required any surgical intervention; that means a 28.5% rate of conservative management for simple MI). 

The newborn needs to undergo fluid resuscitation (150 mL/kg/day minimum) via an intra-venous (IV) line prior to contrast because the hypertonicity of the enema might cause severe fluid changes, cardiovascular collapse, and even end-organ damage including necrotizing enterocolitis [3]. Up to 2–3 gentle enemas with hypertonic solution or acetylcysteine can be performed, under imaging control, with the visualization of the filling of the distal ileum, followed by the evacuation of meconium, as Noblett described since 1969 [24]. The perforation risk is between 2.7% and 23%; therefore, in order to react appropriately to complications like the necessity for urgent surgical intervention, the presence of an IV line is imperative [3]. In spite of this evidence, the agents that are currently used more frequently, at least in the United States, are Omnipaque (240–350 mOsm/kg water) and Cysto-contray II (400 mOsm/kg water), which are significantly less toxic and less hyperosmolar than Gastrografin [25]. Successful conservative management is associated with a shorter time to full feeds—6 (2–10) days in contrast with 15 days (9–19) in those with laparotomy [8]. In a recently published prospective study on 56 neonates with MI and CF, there was significant variation in the number of enemas per infant (1–4), as well as the concentration of the contrast agents. The mortality rate was 4% [8]. The UK and Ireland have highly diverse and non-standardized early management practices for newborns with MI [8]. Anyway, this was the only study reporting in detail the number and the type of hypertonic solution used for enema, and the results cannot be extrapolated due to the small number of cases (12 cases) [8]. After disimpaction of the terminal ileum with Gastrografin, Noblett used to recommend the administration of 5 mL of 10% N-Acetylcysteine every 6 h through the nasogastric tube, but that could cause chemical aspiration pneumonia. That is why some authors only recommend the continuation of enemas with saline solution, under serial radiological control [3,17]. 

If the enema maneuver fails or the patient’s condition worsens, then surgical intervention will be performed through mini-laparotomy and resection of the impacted ileum with primary anastomosis or the creation of an ileostomy proximal to the impacted area with dehydrated meconium will be performed (Figure 4 and Figure 5).

The exact moment of the surgical intervention, the type of stoma, the postoperative management, or the optimal moment of closing the stoma varies depending on the case; there is no consensus in these respects. The Bishop–Koop distal stoma with a proximal end-to-side anastomosis, Santulli proximal stoma with a side-to-end distal anastomosis, or Mikulicz double barrel enterostomy are examples of traditional options for creating an ostomy [1,2,16]. Other therapeutic alternatives include enterotomy with evacuation of meconium and intraoperative saline or acetylcysteine irrigations, continued postoperatively through one of the mentioned stomas, or appendicostomy with saline irrigations intra and postoperatively [26]. Also, a minimal enterotomy with the insertion of a T-tube is sometimes recommended, the irrigations on the tube continuing postoperatively until a normal intestinal transit is obtained, at which point the T-tube is removed and the digestive fistula closes spontaneously. If the terminal ileum is blocked with meconium pellets that cannot be detached without significant damage, then resection of the respective area and primary anastomosis or ileostomy are recommended [26].

Boczar et al. in 2015 retrospectively studied eight patients with simple MI and two with complicated MI. Five from eight patients with simple MI received enemas with no success and all received a Bishop–Koop stoma. Their conclusion was that the Bishop–Koop stoma, permitting the passage through the whole gastrointestinal tract, is a safe option; there were no abnormalities related to the electrolyte balance, excessive fluid loss, or body weight deficiencies associated with the stoma [9].

#### 3.2.2. Complicated Meconium Ileus

Intestinal atresia, prenatal volvulus with gangrene or perforation of the dilated intestinal loops which can lead to a meconium cyst, and meconium peritonitis are the characteristics of complicated meconium ileus [3,27]. The clinical presentation is influenced by the timing of the perforation. There is a chance that some meconium may be reabsorbed in the peritoneum prior to birth if perforation happens earlier in pregnancy, leaving only a few calcifications seen on ultrasound or abdominal X-ray after birth. A higher likelihood of meconium peritonitis or meconium cyst is observed if necrosis and perforation happen near to delivery. In a recent study, about 40% of all patients with MI had the complex form; of these, 13% (or one-third of all problematic cases) had intestinal atresia, and more than half of them also had intestinal volvulus. Given that between 8 and 11% of children with ileal atresia also have a CF diagnosis, these data lend support to the recent demand for a CF-oriented diagnostic work-up in any infant presenting with volvulus or jejunal/ileal atresia [13].

The clinical presentation is usually more spectacular than in the case of simple MI, with bilious vomiting, marked abdominal distension, abdominal wall edema, fever (Figure 6). Respiratory distress may also be caused by substantial abdominal distention. If the neonate is stable, a diagnostic contrast enema can help identify malrotation by pinpointing the cecum’s location and identify a microcolon caused by proximal obstruction in the terminal ileum. If the patient’s clinical condition is deteriorated, a simple abdominal X-ray (Figure 7) and an ultrasound are sufficient to guide operative management. Anyway, infants with complicated MI will always require surgery [8].

Free intraperitoneal fluid with floating echogenic particles, single or numerous pseudocysts, collapsed bowel loops alternating with dilated intestinal loops, and hepatic or splenic calcifications can be seen on the postnatal ultrasound of a patient with complicated MI [28]. Caro-Domínguez et al. in 2018, in a retrospective comparative study, wanted to find out the role of postnatal radiographic and sonographic findings in predicting the need for surgery in neonates with MI. They demonstrated that diffuse peritoneal calcification as an isolated finding can be successfully treated non-operatively and imaging findings that predicted the need for surgery were intestinal obstruction, ascites, volvulus, and pneumoperitoneum [14]. 

Clinical presentation of a complicate MI may be a meconium pseudocyst with a fibrous wall with or without calcifications and the loops situated peripheral and usually posterior to the cyst. There are also possible thick vascular adhesions dotted with calcifications. Another form of presentation can be meconial ascites, which can become infected after birth [29]. These cases require surgical intervention to evacuate the ascites or pseudocyst, with the reduction of abdominal pressure. The membrane of the pseudocyst must be excised, this being sometimes difficult because it is very adherent to the intestinal loops (Figure 8). Once more, in order to remove the obstruction and provide continuous irrigation after surgery, an ostomy must usually be created. Padoan et al. demonstrated that a severe post-surgical clinical picture is a risk factor for negative 12-month clinical outcomes in MI-CF newborns; 18% of operated newborns had to undergo further abdominal surgery during the first month of life, for complications [13]. According to other reports, between 20% and 38% of patients with complicated MI will require reoperation for various complications [8,30].

Nevertheless, no comprehensive research has been conducted so far to determine the ideal surgical protocol for MI [17]. A proximal, distal, or Mikulicz double-barreled ileostomy; T-tube enterostomy; appendicostomy for irrigation and evacuation of impacted meconium; or resection of terminal ileon with primary anastomosis or with ileostomy may all be attempted. We found only three studies that reported the types of stomas performed and another study comparing the outcomes of patients depending on the surgical management [8,10,15,16]. 

In the study conducted by Long et al., of the babies having laparotomies, 24 (56%) had a stoma formation; the type of stoma was known in 13 infants: 10 had Mikulicz double-barreled ileostomy, one newborn had a ‘T-tube’ ileostomy, and two had loop ileostomy. No infant was known to have a Bishop–Koop or a Santulli stoma. Nine infants from 24 who had stomas (38%) had complications and the mortality rate was 4%. Anyway, there was no comparation between the outcome of the patients depending on the type of stoma [8].

Askarpour et al. in 2020 performed a retrospective analysis to determine the surgical outcomes of neonates with meconium ileus who underwent Santulli ileostomy compared to cases submitted to loop ileostomy. They analyzed 28 Santulli ileostomy and 23 loop ileostomy cases and found that necrosis, anastomotic leak, and adhesive intestinal obstruction did not differ between the groups. Skin excoriation, ostomy prolapsed, and surgical site infection was significantly lower in the Santulli ileostomy group as compared to the loop group. Ileostomy output in the first week and fourth week was significantly lower in the Santulli group; also, hospitalization in the Santulli group was 12 ± 2.34 days and in the loop ileostomy group was 14.24 ± 1.47 day (*p* < 0.001). The Santulli ileostomy procedure has advantages over loop ileostomy in terms of postoperative outcomes, providing the finest esthetic results with the least amount of problems [16]. 

On the other hand, Farrelly et al. in 2014 analyzed 39 newborns with divided stomas, 7 with enterotomy and washout, and 8 with resection and primary anastomosis. They concluded that compared to therapy with stomas, retaining intestinal continuity at the initial laparotomy for MI appears to be safe, associated with shorter hospital stays, and does not increase the rate of unplanned re-laparotomy [10]. 

Jessula et al., comparing a small lot of patients managed with and without ostomies (six patients had enterotomy and washout, five patients had resection and primary anastomosis, and six patients had resection and ostomy), found that median times on TPN, in NICU, and in hospital were not significantly different [15]. 

Continuous fluid resuscitation is part of postoperative care, as well as oral feedings accompanied by vitamins and pancreatic enzyme supplementation once the dehydrated meconium was eliminated. Within 6 to 12 weeks, stomas should be closed to help prevent long-term issues with fluid, electrolyte, and nutritional losses. Due to their higher risk of newborn cholestasis, infants with MI also need to have their liver function checked every week. In addition, hyponatremia can occur in CF patients as a result of increased sweating and intestinal sodium loss [31]. The spot urine sodium to creatinine ratio, which has a goal of 17 to 52 mmol/mmol, is a more sensitive indicator of sodium status. This is especially true in infants who have had substantial bowel resection, whether or not they have an ileostomy, because of the excessive fluid and sodium loss [17]. 

### 3.3. Short- and Long-Term Prognosis

For patients with simple MI, successful conservative management is associated with a shorter time to full feeds, 6 days in contrast with 15 days in those with laparotomy, as demonstrated by Long et al. in 2021 in a prospective population-cohort study [8]. Farelly et al. reported a significant difference in LOS between operative patients managed with and without stomas: 49 versus 23 days [10]. 

When it comes to infants with complex MI, they frequently need TPN and lipids at first to sustain their growth. To reduce the incidence of cholestasis, the lipid selection should, if at all possible, favor an anti-inflammatory profile that includes fish oil and medium chain triglycerides (MCT). Proteins can be more easily digested using formulae based on amino acids or protein hydrolysates, whereas MCT are absorbed straight into portal circulation, avoiding the lymphatic system [32]. Breastmilk or conventional formulae can be offered as soon as the baby’s gastrointestinal tolerance increases and they continue to heal from the initial insult of MI [13]. The combined pathophysiology of exposure to TPN and lipids, nothing *per os* condition encouraging biliary sludge, CFTR malfunction within biliary ducts, and intestinal obstruction are the primary causes of increased liver enzymes and cholestasis. When enteric feedings are introduced and advanced and ursodeoxycholic acid is used, the newborn cholestasis typically resolves entirely within three months. Although there has been evidence of a link between history of MI and CF-associated liver illness later in life, this link is still debatable because different research has produced conflicting results [6,33,34]. Supplementation of the diet with fat-soluble vitamin and pancreatic enzymes should also be carried out in children who start enteral nutrition with formula or breastmilk, starting with 2000–4000 lipase units per 120 mL of formula [35]. To maximize the effectiveness of pancreatic enzymes, acid-suppressing drugs such proton-pump inhibitors (PPIs) or histamine blockers are required to raise the pH of the gastric and proximal duodenal fluid [3].

An overall body sodium deficit may arise from excessive intestinal sodium losses when an ileostomy is present, contributing to metabolic acidosis and poor weight gain [2]. Thus, a supplementation of sodium may be given by increasing sodium in TPN or increasing sodium supplementation in enteral feeds. All these special dietary needs lead to prolonged hospitalization of patients with complex MI. In the study conducted by Jessula et al., median NICU and hospital stays were 34.5 and 70 days while median time on TPN and time to ostomy reversal were 28.5 and 97 days, respectively [15].

Patients with CF and MI have equivalent pulmonary function to those without MI at the ages of 15 and 25, but may have lower height and weight percentiles [2,36]. The relationship between the initial MI presentation and subsequent morbidity and mortality has been investigated in a number of retrospective or prospective studies [10,36,37]. There were no long-term differences between the groups in the retrospective studies that evaluated the nutritional status and pulmonary function of patients with CF and MI versus controls who were diagnosed due to other symptoms [10,36]. A nutritional deficiency was linked to worsening lung function in children and adolescents with cystic fibrosis, but not to morbidity in the prospective study conducted by Huaschild et al. [37].

Anyway, according to other studies on this issue, having MI will still put the patient at a major disadvantage even in this decade, with a significant higher mortality. These studies found growth retardation in patients with CF who present with MI and worse than ideal results in later life, particularly when it comes to a reduced forced expiratory volume in 1 s (FEV_1_) [13,38]. Out of a total of 107 CF patients analyzed over a period of over 18 years, the 25 patients diagnosed with MI had the worst pubertal growth and adult height compared to patients diagnosed later, based on other symptoms [39]. In our study, the mortality rate varied between 0% and 8.1% for the neonates with MI and CF [8,10,12,13].

Regarding the long-term evolution of digestive disorders in patients with cystic fibrosis, MI has previously affected about 50% of distal intestinal obstruction syndrome patients (DIOS) [40]. DIOS is the meconium ileus equivalent, affecting 15 to 20% of children and adults with cystic fibrosis. Farelly et al. showed that presentation with MI did not predispose to later development of DIOS *per se*, but patients who presented with MI had a predisposition to develop a more severe form of DIOS necessitating surgery [10]. Other digestive complications of CF are rectal prolapse and fibrosing colonopathy, with episodes of intestinal occlusion due to stenosis [41] (Figure 9).

Patients diagnosed later, without MI, have the benefit of a less severe disease but the drawback of a delayed diagnosis. Patients with MI may have a more severe disease but benefit from an early diagnosis and treatment. That is why some authors even advise treating every MI patient as though they have CF until CF is ruled out in order to provide the best possible care, which includes starting pancreatic enzyme replacement therapy—a crucial step in the process [42]. 

## 4. Discussions

Cystic fibrosis, also called mucoviscidosis, is one of the most frequent autosomal recessive genetic disease among Caucasians, with an incidence of 1 in 2500 live births [1,41]. If in the previous decades the life expectancy of these patients was 30–40 years, thanks to the discoveries and innovations in the genetic, medical, and technological fields; more aggressive use of antibiotics; and intensive nutritional support, currently the life expectancy exceeds 50 years on average [43]. The first manifestation of this condition can be MI, and this form of neonatal occlusion can endanger the patient’s life. Although these days approximately 80% of simple and severe MI cases have both early and late survival rates regularly documented [3,44], there is still no standardized protocol of diagnostic methods and no consensus on treatment methods. In 2014, Smyth et al. published the “European Cystic Fibrosis Society Standards of Care Best Practice guidelines”, but the chapter “What is the best way to manage meconium ileus (MI) in patients with CF?” only has a few lines mentioning that both surgical and non-surgical management should be known to the surgical team; a complicated MI may need longer hospital stays, be more severe, and be more challenging to treat; it could be necessary to use a facility experienced in the dietary management of small bowel syndrome for postoperative care [45].

This is the only systematic review in the last 20 years regarding the diagnosis and treatment of meconium ileus in cystic fibrosis, trying to find the latest recommendations regarding the diagnosis and treatment of MI in CF. Although the diagnosis can be established prenatally, we only found three original studies addressing the issue of prenatal ultrasound findings and the impact on postnatal management of neonates with MI and CF [12,13,15], and no study about possible prenatal management. Treating moms with CFTR correctors and potentiators to act on aberrant CFTR in the fetus is one possible option to take into consideration. The natural course of MI may be altered by these drugs if they are begun at an early enough stage [3,46,47]. To stop the development of a microcolon or other problems, in utero surgical decompression of the occluded bowel or correction of MI would be another possible prenatal intervention.

Neonates with simple MI may achieve successful decompression after noninvasive management with Gastrografin, other contrast repeated enemas, or with acetylcysteine, but the percentage of babies who are treated without surgery has been declining over time. In the last 10 years, we only found 5 studies reporting the results after conservative treatment of MI, with a mean success rate of 28.5%. These results could be the result of contrast agent use, radiologist experience changes, or unwillingness to repeat enemas. Consequently, increased rates of surgical intervention are currently noted. There is a need for more precise guidelines regarding when to resort to surgical treatments, as there remains a considerable percentage of cases (up to 72%) in which the conservative therapy may not be effective. Surgeons should advise repeat enemas prior to laparotomy in patients who are stable. If the enema maneuver fails or the patient’s condition worsens, then surgical intervention will be performed and a stoma created. Although many reviews discuss the benefits or disadvantages of various types of stomas or resection of the distal ileal loop and primary anastomosis, the only original study from the analyzed period recommends the Bishop–Koop stoma for simple meconium ileus that does not respond to enemas [9]. If simple MI responds to enemas, there is still no consensus on the method of continuing the treatment with acetylcysteine or on the moment when enemas are no longer necessary.

For complicated MI, although alternative techniques have been identified, the two basic categories of strategies are those that involve creating a stoma and those that preserve bowel continuity. Comparing patients managed with and without ostomies, median times on TPN, in NICU, and in hospital were not significantly different in the study conducted by Jessula et al. on 19 patients [15]. Instead, in their 2014 study on 54 patients, Farelly et al. concluded that compared to therapy with stomas, retaining intestinal continuity at the initial laparotomy for MI appears to be safe, associated with shorter hospital stays, and does not increase the rate of unplanned re-laparotomy [10]. They reported an increased incidence of complex MI (49%), without mentioning exactly what type of complex MI the patients had. However, in this way they justified the low rate of use of conservative therapeutic methods (Gastrografin enema successful in only 9%) [10]. Regarding the type of stoma, no conclusion can be drawn from the studies of the last 10 years; it seems anyway that Santulli or Bishop–Koop type stomas have superior results compared to loop stomas in patients with complicated MI [9,16]. For a correct comparison of the data and relevant conclusions, there should be uniformity and a standardization of the diagnosis and the treatment methods.

### Study Limitations

Frequent to most systematic reviews on MI treatments, include the small number of original studies in the last 10 years and small number of analyzed patients (out of 566 publications verified, only 8 met the inclusion criteria). The conclusions reached may not be robust or generalizable due to the small dataset. Most studies were retrospectively conducted which also limits the generalization of the results and there may be a high risk of reporting bias. Conservative or surgical management is not standardized between studies and very little comparative data exist to determine which kind of surgery or stoma is ideal, which make it difficult to harmonize results. Neonates with MI and pediatric patients with CF had promising responses to treatment with CFTR correctors and potentiators to act on aberrant CFTR, but RCTs are needed to comprehensively understand their efficacy. Early diagnosis of CF and the initiation of appropriate, multidisciplinary management of MI may help mitigate the risk of developing irreversible digestive and pulmonary damage that will necessitate invasive surgical interventions in these patients.

## 5. Conclusions

Neonates with cystic fibrosis and meconium ileus are treated in very different ways and according to different national standards. This brings to light a major problem in the subject because it could be difficult to attain firm findings due to the diversity of practices. This study reveals a lack of strong data to support management decisions, unequivocally shows that the care of infants with MI is not standardized, and suggests a great need for international collaborative studies.

Anyway, until proven otherwise, each patient with MI should be treated as though they have CF and receive the proper care, as research has demonstrated that early detection and treatment by a multidisciplinary team improves these patients’ short- and long-term prognosis. When feasible, non-operative management is probably ideal for the baby because it saves surgery and makes the transition to full enteral feedings easier. If laparotomy is needed, ileostomy formation is the most common approach. The prognosis for CF patients with MI is now similar to that of CF patients without MI thanks to the use of contrast enemas as a treatment for simple MI, improved surgical methods, and early adoption of nutritional support for complicated MI.

## Figures and Tables

**Figure 1 diagnostics-14-01179-f001:**
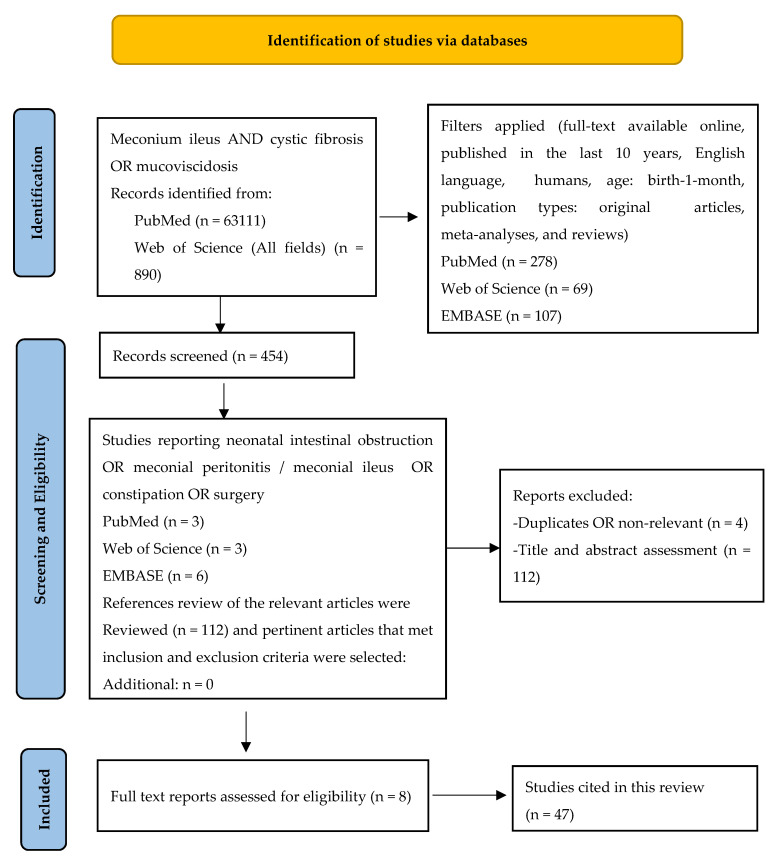
PRISMA flowchart. Notes: PRISMA figure adapted from Liberati A, et al. [11], Creative Commons.

**Figure 2 diagnostics-14-01179-f002:**
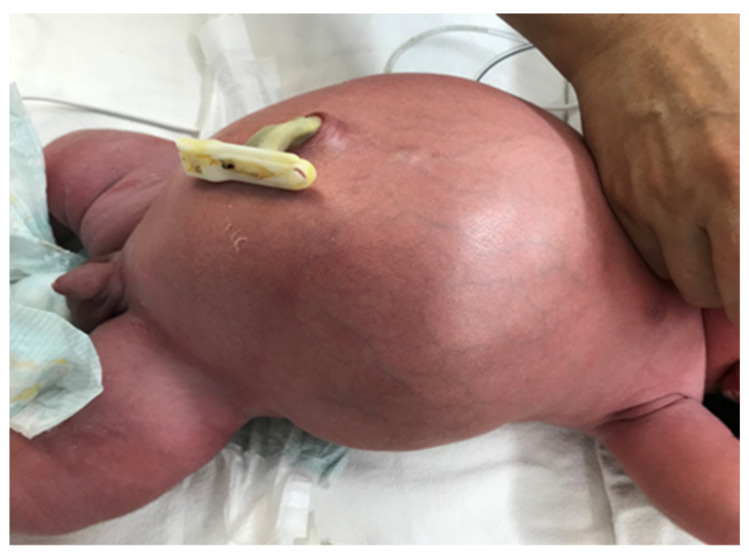
Clinical appearance of a patient with simple meconium ileus.

**Figure 3 diagnostics-14-01179-f003:**
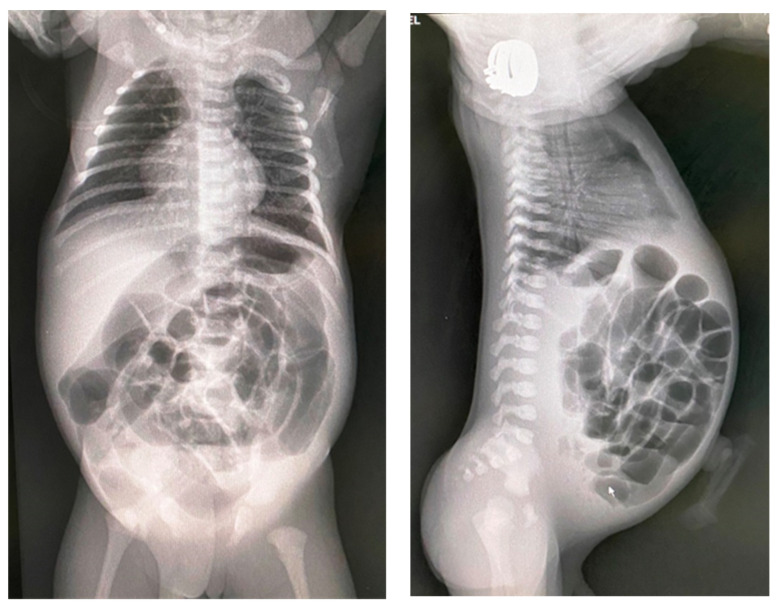
Abdominal X-ray showing dilated intestinal loops and a ground glass appearance; antero-posterior incidence and profile incidence.

**Figure 4 diagnostics-14-01179-f004:**
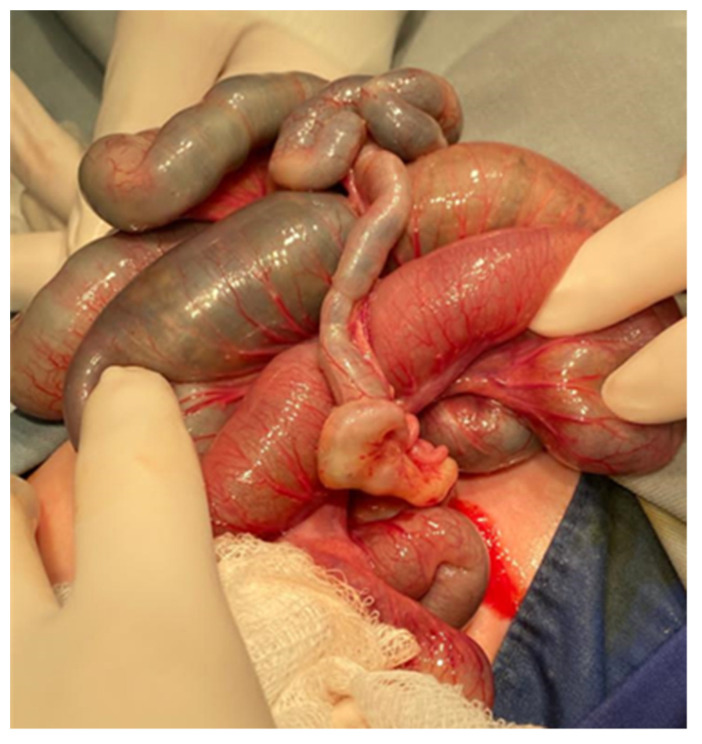
Intraoperative appearance of a meconium ileus showing a mesenteric defect, dilated intestinal loops and a meconium inspissated distal ileum.

**Figure 5 diagnostics-14-01179-f005:**
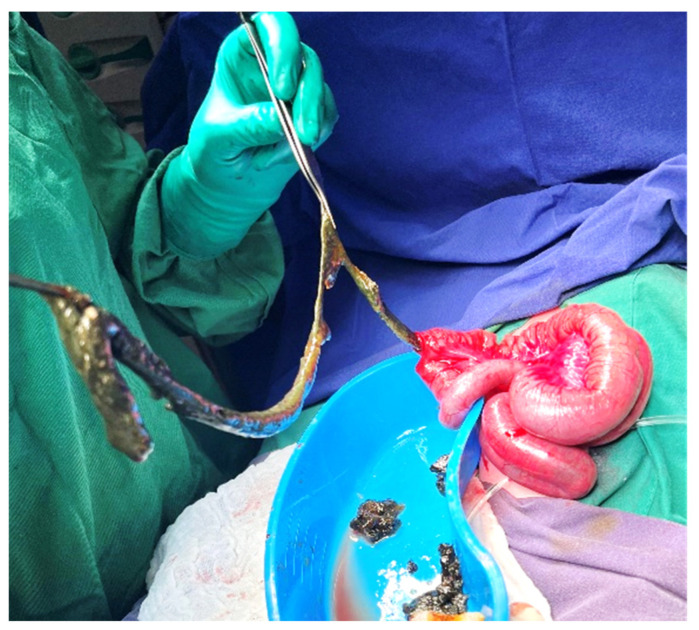
Intraoperative appearance of a dense, adherent meconium extracted from distal ileum.

**Figure 6 diagnostics-14-01179-f006:**
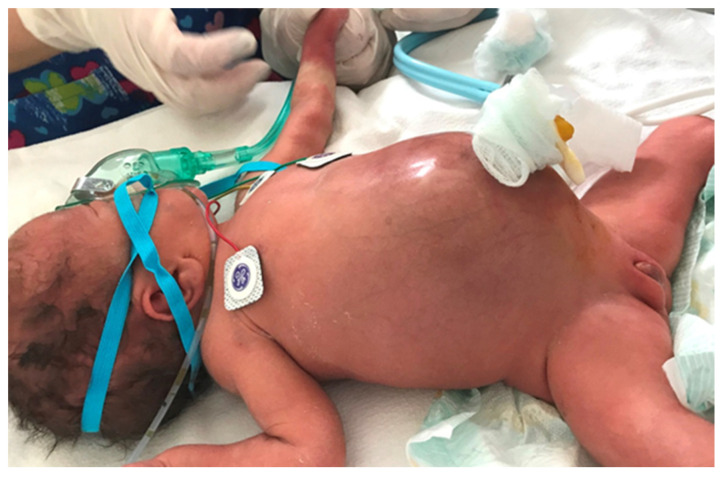
Clinical appearance of a patient with meconium peritonitis.

**Figure 7 diagnostics-14-01179-f007:**
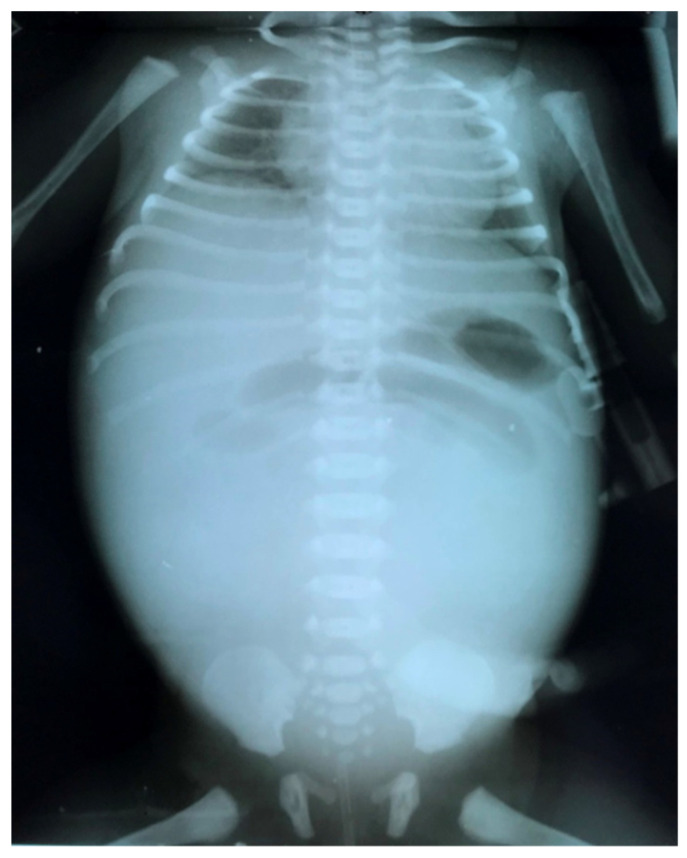
Abdominal X-ray showing dilated intestinal loops in the upper part of the abdomen and opacity due to meconium cyst in the rest of the abdomen.

**Figure 8 diagnostics-14-01179-f008:**
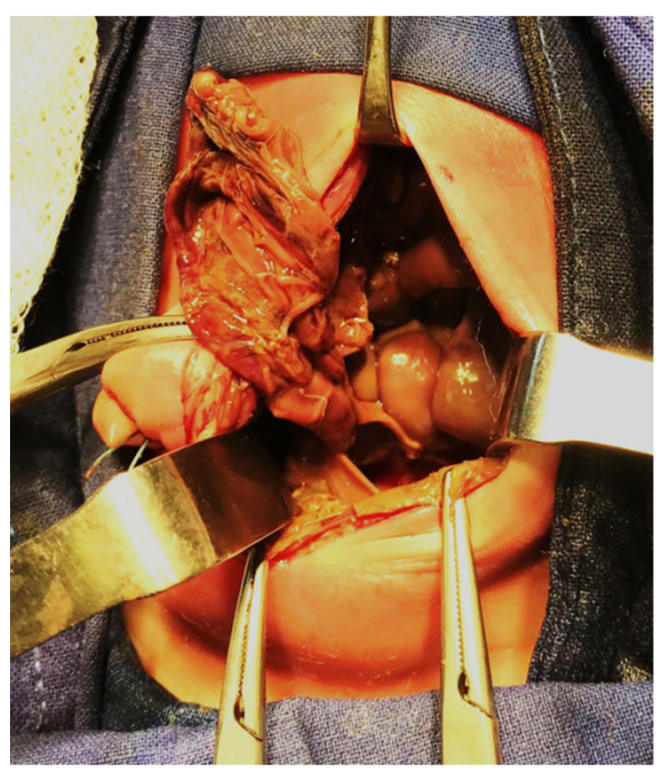
Intraoperative appearance of a meconium peritonitis with removal of the cystic membrane.

**Figure 9 diagnostics-14-01179-f009:**
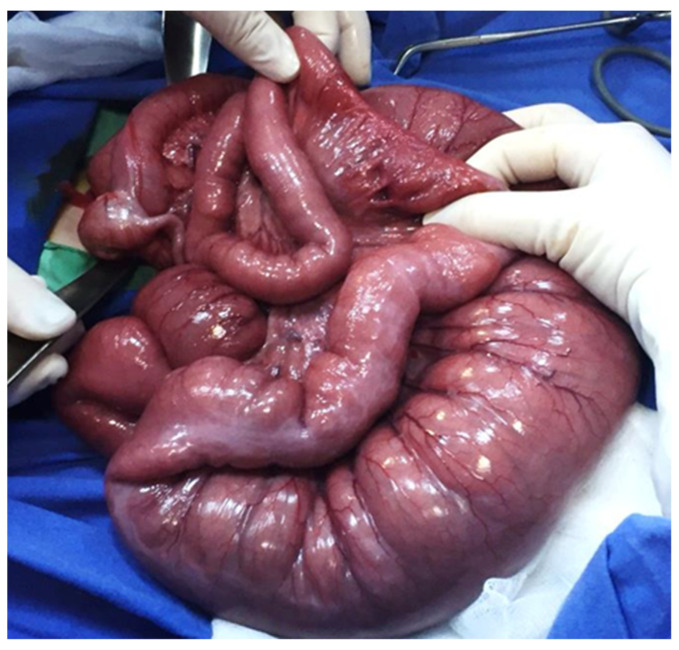
Intestinal stenosis with dilation of the ileon above in a patient with CF.

**Table 1 diagnostics-14-01179-t001:** Procedural treatments for neonates with MI.

Author and Publication Year and Study Design	Purpose of the Study	Simple MI - Enema Treatment	Simple MI Requiring Surgery	Complex MI Requiring Surgery	Reported Outcomes/ Complications/Mortality	Conclusions
- Shinar et al., 2020 [12] - Case Series and Meta-Analysis	Identifying prenatal predictors of neonatal surgery	24 patients with MI and CF, from 244 MI studied patients			Postoperative mortality rate of 8.1% and a survival rate of 100% in neonates not requiring surgery	Meconium pseudocysts, bowel dilation, and ascites are prenatal predictors of neonatal surgery in cases of meconium peritonitis
- Padoan et al., 2019 [13] - Retrospective, multicenter, observational study	Identifying the risk factors for poor 12-month clinical outcomes in MI-CF newborns	24/52 patients treated with enema—success rate 58% (14 cases)	- 38 patients with simple MI - 33 patients with complicated MI needed surgery - 40 subjects underwent intestinal resection - 41 patients needed a stoma		- 31 patients (37%) experienced negative outcomes; the risk factors were prenatally diagnosed intestinal obstruction and a need for intensive care and oxygen therapy - 13/71 patients (18%) had to undergo further abdominal surgery during the first month of life, for complications - Mortality rate 4.7%	High b-IRT levels, prenatally diagnosed intestinal obstruction, a severe post-surgical clinical picture and early liver disease are risk factors for negative outcomes. Breastfeeding may be protective.
- Caro-Domínguez et al., 2018 [14] - Retrospective comparative study	Role of postnatal radiographic and sonographic findings in predicting the need for surgery in neonates with MI	14 patients successfully treated with enema	- 23 patients required surgery (62%) - **Only 8% of patients had CF**		- diffuse peritoneal calcification as an isolated finding can be successfully treated non-operatively	Imaging findings that predicted the need for surgery were intestinal obstruction, ascites, volvulus, and pneumoperitoneum
- Long et al., 2021 [8] - Prospective population-cohort study	Outcome and factors associated with successful non-operative management	12/33 patients were treated with enema alone (36%)	- 20/33 patients were treated with enema and laparotomy (61%) - 1 patient with simple MI was primary operated (3%) - 9/21 (43%) had stoma formation	- 10/23 had enema and laparotomy - 13/23 had primary laparotomy - 15 (65%) had stoma formation	- When using an enema to decompress, infants with simple MI took 6 days to begin complete enteral feedings, whereas those who underwent a laparotomy after an enema took 15 days - Three contrast enemas (5%) were followed by complications: one drop in serum sodium and two intestinal perforations - Nine infants from 24 who had stomas (38%) had complications - Mortality rate was 4%	Compared to those who underwent a laparotomy, infants with simple MI who had successful enema decompression were more likely to have experienced recurrent enemas. Time to full feeding was shortened in cases when non-operative treatment was successful.
- Farrelly et al., 2014 [10] - Retrospective case note analysis	Outcomes of the surgical management for MI and DIOS in CF	19/30 patients with simple MI were treated with enema and only 5 responded	25 patients with simple MI and 29 with complicated MI were operated: - 39 (72%) had divided stomas, 7 (13%) were managed with enterotomy and washout, and 8 (15%) with resection and primary anastomosis		- time to closure for patients managed with stomas was 47 days (4–202) - significant difference in LOS between operative patients managed with and without stomas: 49 versus 23 days - Presentation with MI did not predispose to later development of DIOS *per se* - no postoperative deaths	Compared to therapy with stomas, retaining intestinal continuity at the initial laparotomy for MI appears to be safe, associated with shorter hospital stays, and does not increase the rate of unplanned re-laparotomy - patients who presented with MI had a predisposition to develop a more severe form of DIOS necessitating surgery
- Jessula et al., 2018 [15] - Retrospective cohort review of prospectively collected data	Identifying if specific genetic subtypes and the presence of echogenic bowel on prenatal US are associated with either MI or operative requirement in the neonatal period	7 patients with simple MI were treated conservatively and did not require any surgical intervention	- 3 patients with simple MI and 16 with complicated MI were operated - 2 patients had surgeries performed in another province - 6/19 patients had enterotomy + washout - 5/19 patients had resection and primary anastomosis - 6/19 patients had resection and ostomy - 4/17 patients had a jejunal atresia, 1 patient had duodenal atresia and 1 had gastroschisis		- 6 operated patients presented complications (2 perforations and 4 bowel obstructions) - median age at surgery was 2 days - Median NICU and hospital stays were 34.5 and 70 days while median time on TPN and time to ostomy reversal were 28.5 and 97 days, respectively	- Prematurity and LBW are risk factors for MI and need for surgery. Specific genotypes and echogenic bowel were not predictors of either. - Comparing patients managed with and without ostomies, median times on TPN, in NICU, and in hospital were not significantly different
- Boczar et al., 2015 [9] - Retrospective study	Evaluation of diagnostic and treatment procedures in children with MI	5/8 patients with simple MI received enemas, with no success	- 8 patients with simple MI received a Bishop–Koop stoma and - 2 patients with complicated MI received a stoma - 5 patients were operated on during the first day of life, 4 on the second day and 1 on the third day of life		- The mean hospital stay was 22.9 days - In 8 children the stoma was taken out at the mean age of 19.4 months, in one patient the stoma closed spontaneously	The Bishop–Koop stoma, permitting the passage through the whole gastrointestinal tract, is a safe option - There were no abnormalities related to the electrolyte balance, excessive fluid loss, or body weight deficiencies associated with the stoma.
- Askarpour et al., 2020 [16] - Retrospective analysis	Surgical outcomes of neonates with meconium ileus who underwent Santulli ileostomy were compared to cases submitted to loop ileostomy		- 53 patients with MI received an ostomy - 28 Santulli ileostomy - 23 loop ileostomy		- necrosis, anastomotic leak, adhesive intestinal obstruction did not differ between the groups - skin excoriation, ostomy prolapsed, and surgical site infection was significantly lower in the Santulli ileostomy group compared to the loop group - ileostomy output in the first week and in 4th week was significantly lower in Santulli group - hospitalization in the Santulli group was 12 ± 2.34 days and after loop ileostomy was 14.24 ± 1.47 day (*p* < 0.001)	The Santulli ileostomy procedure has advantages over loop ileostomy in terms of postoperative outcomes, providing the finest esthetic results with the least amount of problems.

MI—meconium ileus; CF—cystic fibrosis; b-IRT—blood immunoreactive trypsinogen; LOS—length of stay; LBW—lower birth weight; US—ultrasound.

## Data Availability

Dataset available on request from the authors.

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
