# Peer review of "Diagnosis and Management of Simple and Complicated Meconium Ileus in Cystic Fibrosis, a Systematic Review"

_diagnostics, 2024, doi:10.3390/diagnostics14111179_

Round 1

Reviewer 1 Report

Comments and Suggestions for Authors

In this systematic review study, Donos et al. examined the diagnosis and management of meconium ileus (MI) in cystic fibrosis patients, highlighting the lack of standardized care and the need for international collaborative studies to establish effective treatment protocols. This systematic review study presents an interesting topic, and the manuscript is well-written. Here are some comments on this review:

1.       Line 55 “HCO3–” and “Ca2+” should be “HCO3” and “Ca2+”.

2.       Section 2.1 recommends that authors indicate the time of the search.

3.       Line 103, please define the abbreviation “TPN”.

4.       It is recommended that the significance of this study be further emphasized and highlighted at the end of the introduction.

5.       Line 281 “The mortality rate was 4%”, need to cite references.

6.       It is suggested that the authors could organize Table 1 more neatly and clearly, such as a three-line table.

7.       It is suggested that the authors indicate the source of the figures used in this study.

Author Response

Dear Reviewer,

Thank you very much for evaluating our manuscript. Your recommendations and comments have helped us improve our manuscript. Here we provide the requested corrections and address the comments. The changes we have made in the manuscript are highlighted in red.

In this systematic review study, Donos et al. examined the diagnosis and management of meconium ileus (MI) in cystic fibrosis patients, highlighting the lack of standardized care and the need for international collaborative studies to establish effective treatment protocols. This systematic review study presents an interesting topic, and the manuscript is well-written. Here are some comments on this review:

  1. Line 55 “HCO3–” and “Ca2+” should be “HCO3–” and “Ca2+”.

Response: We corrected

  1. Section 2.1 recommends that authors indicate the time of the search.

Response: we added the date.

  1. Line 103, please define the abbreviation “TPN”.

Response: We corrected

  1. It is recommended that the significance of this study be further emphasized and highlighted at the end of the introduction.

Response: In lines 70-75 we explained the premises of our study. For a better highlighting of them, we added a phrase at the end of the Introduction chapter.

  1. Line 281 “The mortality rate was 4%”, need to cite references.

Response: We corrected

  1. It is suggested that the authors could organize Table 1 more neatly and clearly, such as a three-line table.

Response: Although we slightly modified table 1, we did not manage to make it only three lines.

  1. It is suggested that the authors indicate the source of the figures used in this study.

Response: All our tables and figures are original. There are no copyright issues. The photos are of our patients, and we have the consent of the parents to use the images for medical and scientific purposes. We added this statement at the end of our manuscript.

Thank you again for reviewing our manuscript.

Reviewer 2 Report

Comments and Suggestions for Authors

After a careful review I have highlighted below a few aspects on the manuscript. This article sets out to explore a difficult topic that completely lacks standardisation, and the authors admit this aspect from the start. The recommendations are general views on the article itself and in my opinion there’s not much the authors can do to improve. Final recommendations at the end.

1.        Lack of Standardization and Consistency in Data:

The abstract clearly states that the management of neonates with meconium ileus (MI) and cystic fibrosis (CF) is highly variable and not standardized across countries. This highlights a significant issue in the field, but also points to a potential weakness in the systematic review itself, as the heterogeneity of practices may complicate the drawing of definitive conclusions.

2.        Limited Scope of Included Studies:

Out of 566 publications verified, only 8 met the inclusion criteria. This small number raises concerns about the comprehensiveness and representativeness of the findings. With such a limited dataset, the conclusions drawn may lack robustness and generalizability.

3.        Contradictory Findings:

The abstract mentions that the conclusions are contradictory, especially regarding the types of stomas performed and their outcomes. This indicates a lack of consensus in the existing literature, which diminishes the strength of any recommendations that the review might make.

4.        Small Sample Sizes:

The studies analysed had small sample sizes, which is explicitly noted as a limitation. Small sample sizes reduce the statistical power of the findings and increase the margin of error, making it difficult to confidently apply these results to the broader population of MI-CF patients.

5.        Limited Comprehensive Research:

The abstract notes that no comprehensive research has been conducted to determine the ideal surgical protocol for complicated MI. This gap in the literature is critical and underscores a major limitation of the review—without comprehensive studies, it is challenging to provide evidence-based recommendations.

6.        High Rates of Complications and Reoperations:

The review found that between 18% and 38% of patients with complicated MI will require reoperation for various complications, and the mortality rate varies between 0% and 8%. These high rates of complications and reoperations suggest that current management strategies are suboptimal, yet the review does not provide clear guidance on how to improve these outcomes due to the lack of robust data.

7.        Need for International Collaborative Studies:

The conclusion calls for international collaborative studies, highlighting a significant gap in current research efforts. While this is a valid recommendation, it also emphasizes that the current state of research is insufficient and fragmented, limiting the practical utility of the review.

8.        Ambiguous Impact of Prenatal Predictors:

While the review identifies prenatal ultrasound findings as predictors of neonatal surgery and negative outcomes, it does not provide clear guidelines on how this information should be used in clinical practice. This ambiguity limits the practical application of these findings for healthcare providers.

9.        Unclear Effectiveness of Conservative Treatments:

The effectiveness of conservative treatments, such as hypertonic solutions enemas, is stated to be effective in more than 25% of cases. However, this leaves a significant proportion of cases (up to 75%) where the treatment may not be effective, indicating a need for clearer guidelines on when to escalate to surgical interventions.

10.   Registration and Reporting Clarity:

Although the protocol was registered in PROSPERO, the abstract does not provide detailed information on the specific criteria for inclusion and exclusion, the quality assessment of the studies, or how data were extracted and analyzed. Greater transparency in these areas would strengthen the credibility of the review.

Overall, while the abstract highlights important issues in the management of MI-CF neonates, the significant limitations in the existing literature, small sample sizes, contradictory findings, and lack of comprehensive research reduce the potential impact and utility of the review’s conclusions. Further detailed and standardized research is crucial for improving clinical outcomes and developing clear management guidelines.

Comments on the Quality of English Language

Minor spellchecking required.

Author Response

Dear Reviewer,

Thank you very much for evaluating our manuscript. Your recommendations and comments have helped us improve our manuscript. Here we provide the requested corrections and address the comments. The changes we have made in the manuscript are highlighted in red.

After a careful review I have highlighted below a few aspects on the manuscript. This article sets out to explore a difficult topic that completely lacks standardisation, and the authors admit this aspect from the start. The recommendations are general views on the article itself and in my opinion there’s not much the authors can do to improve. Final recommendations at the end.

  1. Lack of Standardization and Consistency in Data: The abstract clearly states that the management of neonates with meconium ileus (MI) and cystic fibrosis (CF) is highly variable and not standardized across countries. This highlights a significant issue in the field, but also points to a potential weakness in the systematic review itself, as the heterogeneity of practices may complicate the drawing of definitive conclusions.

Response: As you noted form the Abstract, there is a lack of standardization in the management of children with MI and CF, which makes it difficult to draw relevant conclusions. We emphasized these aspects in the Limitations of the study chapter and also added a sentence regarding this to the Conclusions chapter.

  1. Limited Scope of Included Studies: Out of 566 publications verified, only 8 met the inclusion criteria. This small number raises concerns about the comprehensiveness and representativeness of the findings. With such a limited dataset, the conclusions drawn may lack robustness and generalizability.

Response: Yes, we agree that the conclusions reached may not be robust or generalizable due to the small dataset. We added this phrase to the Limitation of the study section.

  1. Contradictory Findings: The abstract mentions that the conclusions are contradictory, especially regarding the types of stomas performed and their outcomes. This indicates a lack of consensus in the existing literature, which diminishes the strength of any recommendations that the review might make.

Response: Yes, we agree about the lack of consensus and the contradictory findings. We mentioned this in our manuscript.

  1. Small Sample Sizes: The studies analysed had small sample sizes, which is explicitly noted as a limitation. Small sample sizes reduce the statistical power of the findings and increase the margin of error, making it difficult to confidently apply these results to the broader population of MI-CF patients.

Limited Comprehensive Research: The abstract notes that no comprehensive research has been conducted to determine the ideal surgical protocol for complicated MI. This gap in the literature is critical and underscores a major limitation of the review—without comprehensive studies, it is challenging to provide evidence-based recommendations.

High Rates of Complications and Reoperations: The review found that between 18% and 38% of patients with complicated MI will require reoperation for various complications, and the mortality rate varies between 0% and 8%. These high rates of complications and reoperations suggest that current management strategies are suboptimal, yet the review does not provide clear guidance on how to improve these outcomes due to the lack of robust data.

Response: Yes, we agree about these aspects. We mentioned it in the Limitation of the study section.

  1. Need for International Collaborative Studies: The conclusion calls for international collaborative studies, highlighting a significant gap in current research efforts. While this is a valid recommendation, it also emphasizes that the current state of research is insufficient and fragmented, limiting the practical utility of the review.

Response: We stated in the Abstract and in the Conclusion section the need for international collaborative studies.

  1. Ambiguous Impact of Prenatal Predictors: While the review identifies prenatal ultrasound findings as predictors of neonatal surgery and negative outcomes, it does not provide clear guidelines on how this information should be used in clinical practice. This ambiguity limits the practical application of these findings for healthcare providers.

Response: Although the diagnosis can be established prenatally, we only found three original studies addressing the issue of prenatal ultrasound findings and the impact on post-natal management of neonates with MI and CF, and no study about possible prenatal management. Treating moms with CFTR correctors and potentiators to act on aberrant CFTR in the fetus is one possible option to take into consideration. The natural course of MI may be altered by these drugs if they are begun at an early enough stage. To stop the development of a microcolon or other problems, in utero surgical decompression of the occluded bowel or correction of MI would be another possible prenatal intervention.

  1. Unclear Effectiveness of Conservative Treatments: The effectiveness of conservative treatments, such as hypertonic solutions enemas, is stated to be effective in more than 25% of cases. However, this leaves a significant proportion of cases (up to 75%) where the treatment may not be effective, indicating a need for clearer guidelines on when to escalate to surgical interventions.

Response: „Neonates with simple MI may achieve successful decompression after noninvasive management with Gastrografin, other contrast repeated enemas or with acetyl-cysteine, but the percentage of babies who are treated without surgery has been declining over time. In the last 10 years, we only found 5 studies reporting the results after conservative treatment of MI, with a mean success rate of 28,5%. These results could be the result of contrast agent use, radiologist experience changes, or unwilling-ness to repeat enemas. Consequently, increased rates of surgical intervention are currently noted. Surgeons should advise repeat enemas prior to laparotomy in patients who are stable. If the enema maneuver fails or the patient's condition worsens, then surgical intervention will be performed and a stoma created.” – This is stated in our manuscript. We added one phrase to better emphasize the need for clearer guidelines on when to escalate to surgical interventions. Thank you for the comment.

  1. Registration and Reporting Clarity: Although the protocol was registered in PROSPERO, the abstract does not provide detailed information on the specific criteria for inclusion and exclusion, the quality assessment of the studies, or how data were extracted and analyzed. Greater transparency in these areas would strengthen the credibility of the review.

Response: We added in the abstract the inclusion criteria for the selected studies and for the studied patient population.

Thank you again for reviewing our manuscript.